# FMP-AE: A HYBRID APPROACH TO TIME SERIES ANOMALY DETECTION

## ABSTRACT

Unsupervised anomaly detection in time series presents significant challenges, especially due to the lack of labeled data and the prevalence of highly imbalanced datasets. Traditional statistical and machine learning methods often suffer from low recall and computational inefficiency. While deep learning techniques can automatically extract features, they still struggle with data imbalance. This paper introduces a novel anomaly detection model, Feature map Matrix Profile with an AutoEncoder (FMP-AE), which integrates matrix profile techniques with deep learning. The model uses a 1D-CNN to extract features and compute the matrix profile. A new Matrix Profile loss function is introduced and combined with the Autoencoder's reconstruction loss to enhance anomaly detection. The approach also incorporates a sliding window technique to improve sensitivity to sparse anomalies and increase efficiency. Experimental results on the UCR250 benchmark datasets demonstrate the model's superior performance across multiple metrics, including accuracy, precision, recall, F1-score, and AUC. These results highlight the FMP-AE model's ability to efficiently process large-scale datasets and generalize well across diverse time series domains, offering significant improvements in both detection accuracy and computational efficiency.

## 1 INTRODUCTION

As society and industrial processes continue to digitize, numerous sensor-equipped devices generate vast amounts of time series data, including data from financial markets, meteorological data, web traffic monitoring, and manufacturing sensors. Anomaly detection (AD), often referred to as outlier detection (Kieu et al., 2019), is the technique used to identify data points that exhibit substantial deviations from the norm. This field of research has gained widespread attention since the 1960s (Grubbs, 1969), with increasing importance due to growing demand and applications (Pang et al., 2021). The primary goal is to detect these abnormal behaviors to enable corrective or preventive measures. Anomaly detection plays a crucial role across numerous fields, including identifying fraudulent or manipulative trading behaviors in finance (Hilal et al., 2022), monitoring equipment performance to prevent breakdowns in industrial manufacturing (Hsieh et al., 2019), assessing patient vital signs for early detection of potential health problems in healthcare (Chauhan & Vig, 2015), and recognizing unusual network traffic to guard against cyber attacks and data breaches in network security (Hwang et al., 2020). Generally, Anomalies are categorized into three primary categories: point anomalies, contextual anomalies, and collective anomalies, which are common across different datasets.

Time-Series Anomaly Detection (TSAD) is critical in applications like fault detection in industrial systems, health monitoring, and fraud detection in finance. However, TSAD faces unique challenges due to the sequential nature of time-series data. One major issue is the **scarcity of labeled anomalies**, as they are rare and often require domain expertise to identify, making **unsupervised methods** particularly important. Additionally, time-series data often varies significantly across different domains, and existing methods struggle to generalize due to the dynamic temporal dependencies, making **generalizability** a key requirement. Another challenge is **efficiency**. Real-time monitoring systems must process vast amounts of time-series data at scale, yet many current methods lack the computational efficiency needed for large-scale deployment. Despite the development of various approaches, many still depend heavily on labeled data, lack cross-domain adaptability, or are computationally expensive, limiting their applicability in real-world TSAD scenarios.

To overcome these challenges, we propose a novel model, the **F**eature map **M**atrix **P**rofile combined with an **A**utoencoder (**FMP-AE**), which addresses the key issues of label scarcity, generalizability, and computational efficiency. Our approach introduces a new Matrix Profile (MP) loss function that complements the Autoencoder's reconstruction loss, enabling unsupervised anomaly detection without the need for labeled data. This effectively addresses the challenge of label scarcity by leveraging the power of unsupervised learning, allowing the model to detect anomalies in a wide range of time-series data. Additionally, we incorporate a sliding window technique that further mitigates label scarcity and data imbalance by treating smaller subsequences as individual units, making it easier to detect local anomalies and reducing the impact of rare events. To further enhance the generalizability of our model, we conduct experiments on univariate time-series datasets from multiple domains, demonstrating that the FMP-AE model exhibits strong cross-domain adaptability and can generalize well across varied time-series patterns. Additionally, we incorporate 1D Convolutional Neural Networks (CNN) within the FMP-AE model, which not only speeds up Matrix Profile computation but also allows for parallel processing, making the model highly scalable and suitable for large-scale, real-time applications. This design addresses the challenge of efficiency by reducing computational overhead and improving processing time without sacrificing accuracy.

The key contributions of this paper are the following:

- We propose a novel loss function that integrates Matrix Profile (MP) loss with Autoencoder reconstruction loss, addressing the label scarcity challenge by enabling unsupervised learning for time-series anomaly detection.

- We leverage 1D-CNN for feature extraction and Matrix Profile computation, improving computational efficiency while maintaining high accuracy and generalizability across diverse domains.

- We employ a sliding window technique to address the issue of label scarcity and data imbalance, allowing for the detection of local anomalies in imbalanced datasets.

- We conducted comprehensive experiments on the UCR250 benchmark datasets (Wu & Keogh, 2021), which consist of 250 time series from diverse domains. Experimental results demonstrate the superior performance of our model across multiple evaluation metrics. Additionally, we perform five ablation studies to validate the effectiveness of each key component of our approach.

## 2 RELATED WORK

Unsupervised anomaly detection in time series is an essential real-world problem that has been extensively studied. Approaches for detecting anomalies can be categorized into statistical, machine learning, and deep learning methods. Statistical methods detect anomalies by calculating statistical characteristics of time series data. While these methods are generally simple and computationally efficient, they tend to be less effective in capturing complex time series patterns and struggle with non-linear or high-dimensional data. Machine learning methods model time series data using supervised or unsupervised learning algorithms to detect anomalies, such as Local Outlier Factor (Breunig et al., 2000), One-Class SVM (Erfani et al., 2016), and Support Vector Data Description (Zhou et al., 2021). These classic methods do not take into account temporal information, making them challenging to generalize to unknown real-world scenarios.

Deep learning methods have demonstrated considerable advantages in the field of time series anomaly detection in recent years. It aims to learn feature representations and calculate anomaly scores through neural networks for detecting anomalies. Deep anomaly detection methods include Autoencoder (AE) (Sakurada & Yairi, 2014), Variational Autoencoder (VAE) (An & Cho, 2015), Recurrent Neural Networks (RNN), and Long Short-Term Memory networks (LSTM) (Malhotra et al., 2015). In addition, many hybrid approaches have been proposed. The LSTM-VAE method (Lin et al., 2020) uses a VAE to extract robust local features from short windows of the sequence, then pass these features to an LSTM to estimate the long-term dependencies in the sequence, thereby achieving anomaly detection. The Beat-GAN (Zhou et al., 2019) uses the Generative Adversarial Networks to generate samples similar to real data through adversarial training between the generator and discriminator models, detecting anomalies by comparing generated samples with real data.

Anomaly Transformer method (Xu et al., 2022) introduces the Association Discrepancy of time series with a new Anomaly-Attention mechanism to extract information for anomaly detection.

One-dimensional Convolutional Neural Networks (1D-CNN) have been applied to time series data (Yin et al., 2020), utilizing convolutional operations to effectively extract local patterns and features. This approach has proven to be well-suited for detecting local anomalies in time series data, as shown in previous studies. In 2016, Eamonn Keogh's team at the University of California, Riverside, introduced the matrix profile data structure (Yeh et al., 2016), specifically designed to analyze internal associations within subsequences of time series data by computing a similarity matrix. It is primarily used for pattern (i.e., Motif) discovery and anomaly detection in both univariate and multivariate time series data. There are several algorithms for computing Matrix Profile, and efficiency is crucial for large time series. Notable methods include STAMP (Yeh et al., 2016), STOMP (Zhu et al., 2016), and SCRIMP (Zhu et al., 2018). Among them, SCRIMP and anomaly detection techniques like MERLIN++ (Nakamura et al., 2023) achieve the highest accuracy. However, MP methods can be computationally demanding due to pairwise distance calculations. In scenarios where rapid anomaly detection is critical, we aim to optimize the MP computation to reduce costs.

Despite the development of numerous anomaly detection methods for various datasets, a recent survey (Goswami et al., 2023) found that no single method consistently outperforms others across all datasets. They performed extensive experiments on various real-world datasets and introduced an innovative robust rank aggregation approach to merge several surrogate metrics into a cohesive model selection standard.

This study presents an accurate, efficient anomaly detection method with strong generalization ability, demonstrating its effectiveness on the UCR datasets. This approach leverages Autoencoder to capture normal patterns and employs matrix profiles to assess local similarities, significantly enhancing the performance of anomaly detection models.

## 3 Method

A time series $\mathcal{T} = \langle \mathbf{t}_1, \mathbf{t}_2, \ldots, \mathbf{t}_n \rangle$ represents a sequence of vectors arranged in chronological order. A subsequence of length $k$ that begins at position $i$ is represented as $x_i = \langle \mathbf{t}_i, \mathbf{t}_{i+1}, \ldots, \mathbf{t}_{i+k-1} \rangle$, with the condition $1 \leq i \leq n-m+1$. An anomaly within a time series is characterized by a notable divergence in behavior or patterns compared to the series' typical behavior.

Anomaly Score (AS) quantifies how much a data point or vector deviates from the expected normal behavior. For a time series $\mathcal{T} = \langle \mathbf{t}_1, \mathbf{t}_2, \ldots, \mathbf{t}_n \rangle$, the anomaly score $AS(\mathbf{t}_i)$ for each $\mathbf{t}_i$ indicates the likelihood of $\mathbf{t}_i$ being an outlier. Generally, a higher value of $AS(\mathbf{t}_i)$ indicates an increased probability that the vector is an anomaly.

**Problem Description.** For a given time series $\mathcal{T}$, the goal is to develop a model that generates an anomaly score $AS(\mathbf{t}_i)$ for every data point $\mathbf{t}_i$. A higher score indicates a greater deviation from the normal behavior, thus increasing the probability that $\mathbf{t}_i$ is an anomaly.

### 3.1 Calculate optimized-MP by 1D-CNN

One-Dimension Convolutional Neural Network (1D-CNN) is usually used for processing sequence and time series data. Our model framework is shown in Figure 1. First, We preprocess the time series data using a sliding window technique, where the window moves step-by-step across the series, generating overlapping segments. This approach helps handle long sequences and addresses data imbalance and label scarcity, as each segment can be independently analyzed for anomalies. Additionally, we introduce a rule that if any point within a window is flagged as anomalous, the entire window is considered anomalous. This increases the likelihood of detecting larger anomalous regions and ensures that sparse anomalies or larger abnormal events are captured as a whole, rather than fragmented.

Each segment is input into the 1D-CNN for feature extraction. The CNN consists of three convolutional layers with batch normalization and ReLU activation, followed by Max Pooling. The convolutional layers capture local patterns, while pooling reduces dimensionality and highlights key features for anomaly detection. After feature extraction, we compute the Matrix Profile by calculating Euclidean distances between feature maps of different segments. These distances form an

optimized matrix profile, which identifies the most similar segments. The segments with unusually large distances from their most similar counterparts are flagged as anomalies.

The Matrix Profile (MP) serves as a data structure specifically designed for analyzing time series. It consists of a vector that holds similarity scores among the subsequences within the time series. Traditional Matrix Profile is computed directly on raw time series data, but it can be sensitive to noise, trends, and seasonality. In our approach, we first extract high-dimensional feature representations from the time series using a 1D-CNN. These features are more robust and better suited for capturing complex patterns, which allows for a more reliable matrix profile calculation. Given a time series $\mathcal{T}$ and a predetermined subsequence length $k$, the set of all subsequences of length $k$ can be defined as:

$$\mathcal{X}^{(k)} = \{x_i \mid x_i = [\mathbf{t}_i, \mathbf{t}_{i+1}, \ldots, \mathbf{t}_{i+k-1}], i = 1, 2, \ldots, n - k + 1\}$$

where each subsequence $x_i$ is a contiguous segment of length $k$ starting at position $i$ in the time series $\mathcal{T}$. Each segment of the time series is passed through a 1D-CNN to extract local features, as described in the model architecture. These features are more abstract and stable compared to the raw time series data.

Firstly, for each feature vector $\mathbf{f}_i$ extracted from CNN, we calculate the L2 norm:

$$\|f_i\|_2 = \sqrt{\sum_{j=1}^{d} f_{ij}^2} \tag{1}$$

where $d$ is the dimensionality of the feature vector for each segment. Then, we compute the similarity matrix. We calculate the similarity matrix $S$ based on the dot product of feature vectors and normalize it:

$$S_{ij} = f_i \cdot f_j \quad S'_{ij} = \frac{S_{ij}}{\|f_i\|_2 \|f_j\|_2} \tag{2}$$

We use a sliding window of size $k$ to compute the local mean for each window:

$$\mathrm{MP}_i = \frac{1}{k^2} \sum_{p=i}^{i+k-1} \sum_{q=i}^{i+k-1} S'_{pq} \tag{3}$$

Fianlly, we aggregate the local means to construct the optimized matrix profile:

$$\mathbf{P} = [\mathrm{MP}_1, \mathrm{MP}_2, \ldots] \tag{4}$$

Each element of $P$ represents the similarity score for a given subsequence, with lower values indicating high similarity (recurring patterns or motifs) and higher values suggesting anomalies. By using the CNN-extracted features, our approach improves the robustness of the matrix profile calculation. Unlike traditional methods that operate directly on raw data and are sensitive to noise or trends, our model focuses on stable, high-level feature representations. As a result, the optimized matrix profile is better suited for identifying anomalies by highlighting the dissimilarity between subsequences with higher accuracy. Typically, lower values in the optimized MP correspond to recurring patterns (Motifs), while higher values indicate anomalies.

## 3.2 TRAINING FMP-AE WITH NEW LOSS FUNCTION

Given a time series $\mathcal{T}$, we first apply a sliding window of length $k$ to the sequence, generating $m$ subsequences, where $m$ is the number of windows extracted. These subsequences are then passed through a 1D-CNN to extract feature maps. We subsequently compute the Matrix Profile (*MP*) based on these feature maps, with the values in the MP denoted as $p_i = MP[i]$.

The reconstruction loss is defined as the mean squared error (MSE) between the original input sequence $\mathbf{X}$ and the reconstructed output $\hat{\mathbf{X}}$ from the Autoencoder, and the Matrix Profile loss (*MP* loss) as the mean of the elements in the Matrix Profile:

$$\mathcal{L}_{\mathrm{recon}} = \frac{1}{m} \sum_{i=1}^{m} \|x_i - \hat{x}_i\|^2, \quad \mathcal{L}_{\mathrm{MP}} = \frac{1}{m} \sum_{j=1}^{m} p_j, \tag{5}$$

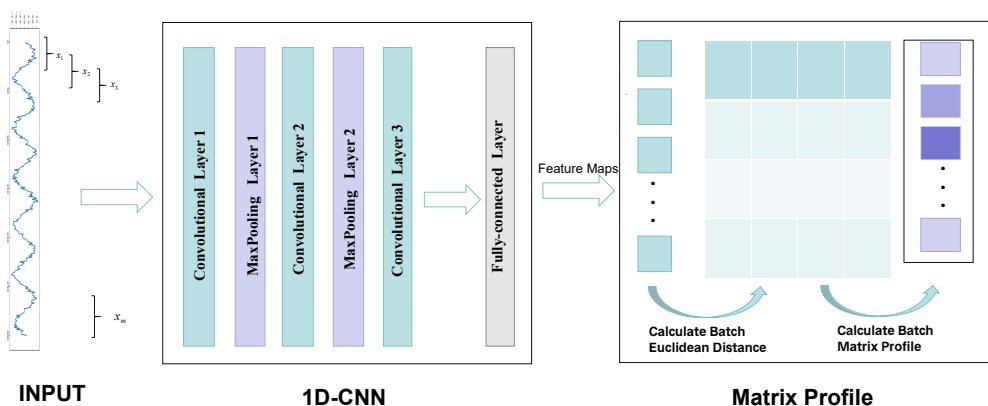

Figure 1: Overview of our proposed model

To ensure consistency across all batches, we normalize both the reconstruction loss and the Matrix Profile loss using global normalization factors computed across the entire dataset. The normalized *MP* loss is defined as:

$$\tilde{\mathcal{L}}_{\text{MP}} = \frac{\mathcal{L}_{\text{MP}}}{\overline{p}}, \quad \text{where} \quad \overline{p} = \frac{1}{N}\sum_{b=1}^{N}\frac{1}{m}\sum_{j=1}^{m}p_j^{(b)} \tag{6}$$

where $N$ is the total number of batches, $p_j^{(b)}$ represents the Matrix Profile value of the $j$-th element in the $b$-th batch.

Similarly, the normalized reconstruction loss is defined as:

$$\tilde{\mathcal{L}}_{\text{recon}} = \frac{\mathcal{L}_{\text{recon}}}{\overline{e}}, \quad \text{where} \quad \overline{e} = \frac{1}{N}\sum_{b=1}^{N}\frac{1}{m}\sum_{i=1}^{m}\|x_i^{(b)} - \hat{x}_i^{(b)}\|^2 \tag{7}$$

where $e_i^{(b)} = \|x_i^{(b)} - \hat{x}_i^{(b)}\|^2$ represents the reconstruction error of the $i$-th element in the $b$-th batch, and $\overline{e}$ is the mean reconstruction error across all batches.

The final total loss function is then defined as:

$$\mathcal{L}_{\text{total}} = \tilde{\mathcal{L}}_{\text{recon}} + \lambda \cdot \tilde{\mathcal{L}}_{\text{MP}} \tag{8}$$

Here, $\mathcal{L}_{\text{recon}}$ measures the Autoencoder's ability to reconstruct the sequence, with higher reconstruction errors indicating potential anomalies. Conversely, $\mathcal{L}_{\text{MP}}$ assesses the similarity between subsequences in the Matrix Profile, where lower similarity (higher MP loss) suggests anomalies. The hyperparameter $\lambda$ balances the contributions of reconstruction loss and MP loss, enabling the model to focus on both reconstruction errors and similarity features.

The method detects anomalies by utilizing both reconstruction error and MP values. A large reconstruction error indicates a potential anomaly, while a high MP value suggests low similarity with other segments, also signaling anomalies. To enhance detection, the weight of the *MP* loss, represented by $\lambda$, is dynamically increased during training. This gradual adjustment is beneficial, as prioritizing local anomaly detection too early may lead the model to converge to local minima or neglect the global data structure. By progressively emphasizing the matrix profile error, the model can initially focus on global features before honing in on local anomaly patterns. Figure 2 illustrates the training process.

The FMP-AE training process involves several steps: First, a 1D-CNN extracts features from data segments, which are then reconstructed using an Autoencoder. The matrix profile is computed from these features, and both reconstruction loss and *MP* loss are integrated into a new loss function. Gradients are calculated through backpropagation, with dynamic adjustments to $\lambda$ to prevent gradient

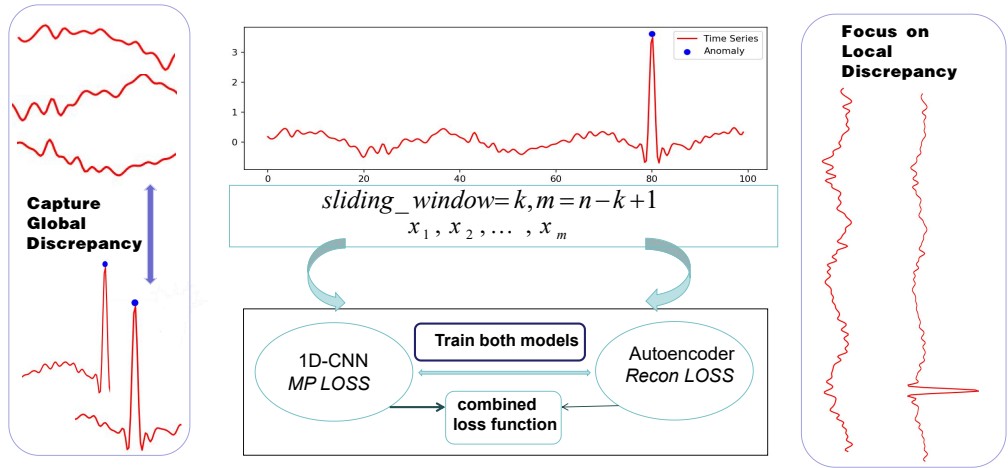

Figure 2: Training model using the combined loss

explosion. As the Matrix Profile evolves during training with updated model parameters, the *MP* loss also changes dynamically. As the model learns data patterns, the *MP* loss gradually decreases, reflecting improved performance in global similarity. This dual learning approach allows the model to capture local features through reconstruction loss while simultaneously enhancing global similarity through *MP* loss, thereby improving anomaly detection effectiveness.

Intuitively, *MP* loss captures internal sequence correlations, while reconstruction loss addresses the overall differences between the original and reconstructed sequences. Reconstruction loss ensures effective reconstruction of normal patterns, while *MP* loss increases sensitivity to anomalies. This combination improves anomaly detection, making the model more accurate and robust in identifying anomaly in time series data.

### 3.3 ANOMALY DETECTION

We propose an integrated Anomaly Score for detecting anomalies. First, we compute the *MP* values $p_i$ based on 1D-CNN. Then we use the Autoencoder to reconstruct the input subsequence $x_i$ and calculate the reconstruction error $e_i = \|x_i - \hat{x}_i\|^2$. Next, we apply Softmax to $p_i$ to get the weights $w_i$:

$$w_i = \frac{\exp(p_i)}{\sum_j \exp(p_j)} \tag{9}$$

The final Anomaly Score $AS_i$ is then computed as $AS_i = w_i \cdot e_i$. This score combines global similarity from the Matrix Profile with local reconstruction errors for effective anomaly detection. Finally we use a threshold $\tau$ to determine anomalies through the following labels:

$$\text{label}_i = \begin{cases} 1 & \text{if } AS_i > \tau, \\ 0 & \text{if } AS_i \leq \tau, \end{cases} \tag{10}$$

the value 1 indicates an anomaly, while the value 0 does not.

## 4 EXPERIMENT

### 4.1 RESULTS AND ANALYSES

**Datasets** The UCR Anomaly Detection datasets (Wu & Keogh, 2021) include 250 files across various domains, such as medicine (64%), biology (22%), industry (9%), and meteorology (5%). Each file contains a training set of normal data and a test set with one anomaly, reflecting the rarity of anomalies in real-world scenarios. Time series lengths range from 6,680 to 900,000 points, with

training sets making up about 31% of the total. All series are min-max normalized. Designed for realism, the UCR datasets address the oversimplification and lack of authenticity in previous datasets.

**Baselines** We compared our FMP-AE model against several baseline methods, including THOC (Shen et al., 2020), OC-SVM (Erfani et al., 2016), IForest (Liu et al., 2008), LOF (Breunig et al., 2000), ARIMA (Yaacob et al., 2010), LSTM-VAE (Lin et al., 2020), LSTM-VAE(p) (Park et al., 2018), OmniAnomaly (Su et al., 2019), BeatGAN (Zhou et al., 2019), ADTransformer (Xu et al., 2022), USAD (Audibert et al., 2020), GDN (Deng & Hooi, 2021), InterFusion (Li et al., 2021), and Deep-SVDD (Zhou et al., 2021). This includes traditional machine learning, deep learning, and hybrid approaches. We evaluated all methods based on Precision, Recall, and F1-score metrics. We have uploaded the code to this link:https://github.com//FMP-AE.

As shown in Table 1, our model outperforms others in both Precision and F1-score, demonstrating a strong balance between Precision and Recall. The visual comparison of the models is displayed in Figure 3.

Table 1: Comparison results between different models

| Model | Precision | Recall | F1-score | Type of paradigm | Category |
|---|---|---|---|---|---|
| THOC | 52.30 | 82.95 | 64.33 | Clustering | Traditional |
| OC-SVM | 41.14 | 90.04 | 57.23 | Clustering | Traditional |
| IForest | 40.77 | 93.60 | 56.07 | Clustering | Traditional |
| LOF | 41.47 | **98.80** | 58.42 | Density-based | Traditional |
| ARIMA | 13.59 | 85.71 | 39.75 | Forecasting | Traditional |
| LSTM-VAE | 65.73 | 89.45 | 75.73 | Reconstruction | Deep |
| LSTM-VAE(p) | 62.08 | 95.54 | 75.89 | Reconstruction | Deep |
| OmniAnomaly | 64.21 | 86.93 | 73.86 | Reconstruction | Deep |
| BeatGAN | 45.20 | 88.42 | 59.82 | Reconstruction | Deep |
| ADTransformer | 72.80 | 99.60 | 84.12 | Reconstruction | Deep |
| USAD | 23.75 | 95.60 | 47.86 | Reconstruction | Deep |
| DCdetector | 61.62 | **100.00** | 74.05 | Reconstruction | Deep |
| GDN | 32.46 | 98.60 | 46.59 | Forecasting Based Graph | Deep |
| InterFusion | 60.74 | 95.20 | 74.16 | Reconstruction | Hybrid |
| Deep-SVDD | 47.08 | 88.91 | 61.56 | Distance-based | Hybrid |
| **Our Model** | **81.03** | 94.30 | **86.79** | Reconstruction | Hybrid |

**Analyses** Among all the evaluated models, our model demonstrates the best overall performance, particularly in terms of precision and F1-score. Our model achieves a precision of 81.03% and a F1-score of 86.79%, showcasing its excellent balance between accuracy and recall. The high precision suggests that our model successfully reduces false positives, while the elevated F1-score demonstrates its effectiveness in maintaining a good balance between precision and recall. In comparison, although the LOF model has the highest recall rate at 98.80%, which suggests it almost never misses any anomalies, its lower precision may lead to more false positives. Other models such as Om-

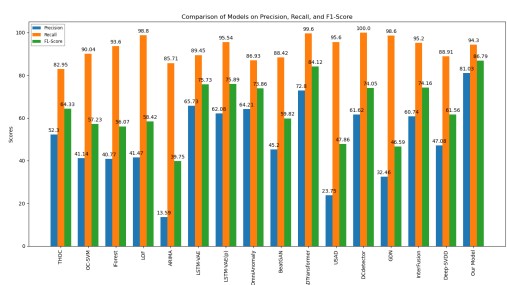

Figure 3: Comparison results of these models

niAnomaly and InterFusion also perform well in terms of precision and recall but do not surpass our model in overall performance. Notably, ADTransformer performs well in both precision and recall, but its F1-score of 82.37% is still lower than our model's 86.79%. This further underscores the exceptional balance our model maintains between accuracy and recall. In summary, our model FMP-AE delivers not only high accuracy in anomaly detection but also achieves an excellent balance between precision and recall, resulting in the best overall performance for the task.

We present the reconstructed loss and *MP* loss in Figure 4. Comparing these two types of loss, we conclude that Reconstruction loss may not effectively distinguish anomaly in some cases, while *MP* loss demonstrates better discrimination ability. The original anomaly map and the detected results (only contain validation data) are shown in Figure 5; the AUC-ROC curve is presented in Figure 6. It can be observed that our model effectively detects anomalies in the time series.

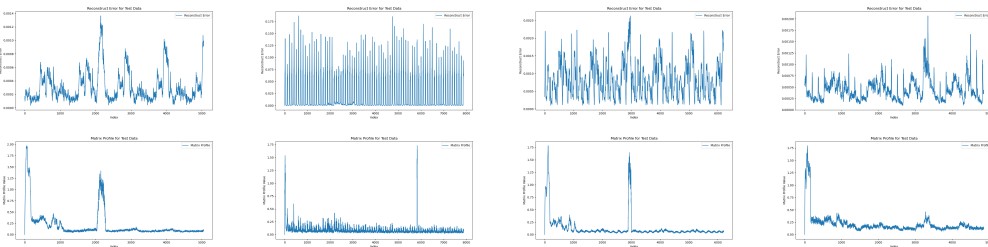

Figure 4: Reconstruction loss (the first line) and *MP* loss (the second line) are plotted alongside the original time series with annotated abnormal regions. For both types of loss, a higher value indicates an anomaly. From the figure, it is evident that *MP* loss demonstrates a stronger ability to identify anomalies compared to reconstruction loss, particularly in aligning with the annotated abnormal regions.

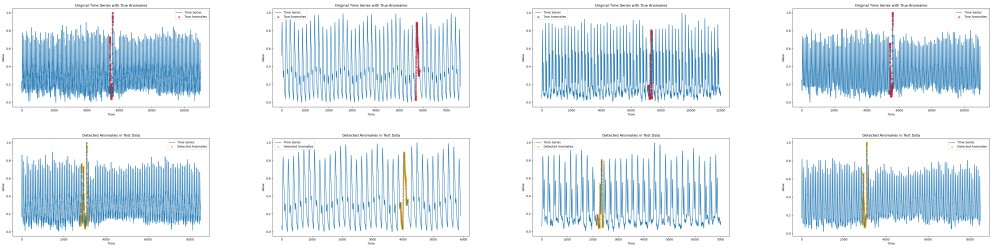

Figure 5: Original time series and detected anomalies (only validation data) are shown on the map. The figure demonstrates how the model identifies anomalies within the time series, highlighting the correspondence between the detected anomalies and the actual abnormal regions. This visual comparison helps in evaluating the effectiveness of the model in anomaly detection on real-world data.

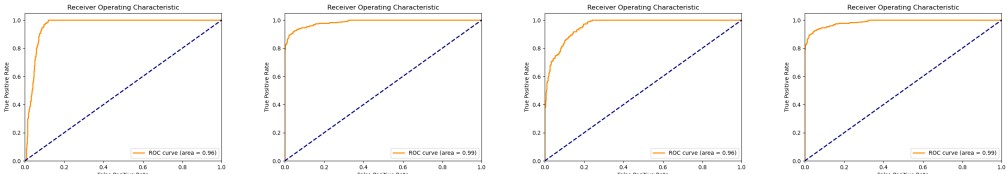

Figure 6: AUC-ROC curve showing the model's performance across various thresholds. A higher AUC value indicates better effectiveness, representing a higher true positive rate and a lower false positive rate. This figure demonstrates the model's overall capability to correctly identify anomalies while minimizing false positives, providing a comprehensive evaluation of its detection performance.

## 4.2 ABLATION EXPERIMENTS

To illustrate the function of each component of FMP-AE, we conducted original experiments (i.e.,MP+1D-CNN+AE) on the UCR datasets and performed five ablation studies, by (1) replacing 1D-CNN with MLP (MP+MLP+AE), (2) removing 1D-CNN feature extraction (MP+AE), (3) removing matrix profile loss (Only AE), (4) removing Autoencoder (MP+1D-CNN) and (5) removing MP (1D-CNN+AE) respectively. The experimental results show the performance of each

experiment in terms of accuracy, precision, recall, and F1-score. The results are presented in Table 2. From these results, we can draw the following conclusions.

Table 2: Performances of five ablations

| Metric | Accuracy | Precision | Recall | F1-score |
|--------|----------|-----------|--------|----------|
| **Our model** | 97.96 | 81.03 | 94.30 | 86.79 |
| MP+MLP+AE | 96.15 | 71.64 | 83.25 | 74.92 |
| Only AE | 96.70 | 60.17 | 84.16 | 67.87 |
| MP+AE | 79.69 | 20.48 | 81.86 | 31.41 |
| 1D-CNN+AE | 90.45 | 65.73 | 73.29 | 69.30 |
| MP+1D-CNN | 89.81 | 27.84 | 89.59 | 40.12 |

It can be seen that our model performs the best across all metrics, maintaining a balance between precision, recall, and F1-score. Based on the experimental results in the Table 2, we can compare each ablation study with the original model (our model) and analyze the performance change.

**MP+MLP+AE (Replacing 1D-CNN with MLP):** Compared to the original model, the accuracy in the MP+MLP+AE experiment dropped from 97.96% to 96.15%, and the F1-score decreased from 86.79% to 74.92%. This decline highlights the critical role of 1D-CNN in extracting local features from time series data. Unlike 1D-CNN, MLP is less effective in capturing temporal dependencies and local patterns, which are crucial for identifying anomalies in time series. This experiment demonstrates that leveraging convolutional operations is essential for enhancing the model's ability to detect subtle anomalies.

**Only AE (Remove *MP* loss):** When using only Autoencoder, the accuracy remains relatively high at 96.70%, but precision drops significantly to 60.17%, and the F1-score decreases to 67.87%. The high recall (84.16%) indicates that Autoencoder alone can detect most anomalies but struggles with precision, leading to more false positives. This result underscores the importance of the MP loss in refining anomaly detection by improving the precision and reducing false positives. It highlights how combining reconstruction-based and pattern-based metrics enables more balanced and reliable anomaly detection.

**MP+AE (Removing 1D-CNN):** After removing the 1D-CNN, the model's accuracy decreases to 79.69%, with a sharp decline in F1-score to 31.41% and precision to 20.48%. This experiment demonstrates that the absence of 1D-CNN significantly weakens the model's ability to extract local features, making it challenging to distinguish between normal and anomalous patterns. The results underline the indispensable role of 1D-CNN in capturing the nuanced temporal structures of time series, which are crucial for robust anomaly detection.

**1D-CNN+AE:** In this experiment, the accuracy drops to 90.45%, with an F1-score of 69.30%. The precision (65.73%) and recall (73.29%) indicate that while 1D-CNN and Autoencoder can still detect anomalies, the absence of MP loss reduces precision and overall performance. This highlights the role of MP loss in refining detection by improving the balance between precision and recall.

**MP+1D-CNN (Removing Autoencoder):** When removing the Autoencoder, the accuracy is 89.81%, and the F1-score is 40.12%. Although the recall remains high at 89.59%, precision drops significantly to 27.84%. This indicates that without the reconstruction-based anomaly detection metric provided by the Autoencoder, the model heavily relies on pattern-based detection, leading to more false positives. This result highlights the complementary nature of the Autoencoder and the MP loss, where the Autoencoder provides a reconstruction-based perspective to better differentiate normal from anomalous patterns.

**AUC-ROC curve:** The performance of the five models is evaluated using Receiver Operating Characteristic (ROC) curves, comparing their True Positive Rate (TPR) and False Positive Rate (FPR). The effectiveness of each model is quantified by the Area Under the Curve (AUC) score, which serves as a comprehensive measure of classification performance. The results are depicted in Figure 7. Among the models, the MP+1D-CNN achieved the highest AUC score, underscoring the pivotal role of 1D-CNN in effectively extracting features for time series anomaly detection. However, despite its superior AUC, the MP+1D-CNN model demonstrates a weaker ability to balance precision and recall compared to our model. Furthermore, substituting 1D-CNN with MLP or employing only

the Autoencoder and Matrix Profile resulted in a marked decline in performance, reinforcing the necessity of 1D-CNN for achieving high accuracy in anomaly detection.

This analysis highlights that, for the specific datasets, 1D-CNN is significantly more effective than MLP or a purely Autoencoder-based approach in capturing local patterns within time series data.

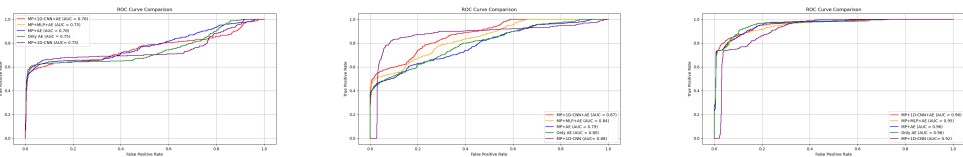

Figure 7: AUC-ROC curve of original model and five ablation experiments

## 5 CONCLUSION AND FUTURE WORK

In this paper, we propose a hybrid method, FMP-AE, which combines traditional matrix profile techniques with deep learning for univariate time series anomaly detection. We employ a 1D-CNN to extract features and compute the matrix profile, while introducing a novel Matrix Profile loss function that integrates with the reconstruction loss during model training. The MP loss captures internal similarities between sequences, and the reconstruction loss accounts for discrepancies between the original and reconstructed sequences. We conducted comparative experiments and five ablation studies on the UCR250 datasets. The results show that FMP-AE significantly outperforms traditional methods in key metrics such as Precision, Recall, F1-score, and AUC-ROC. Our approach not only improves anomaly detection accuracy but also enhances computational efficiency and generalization across diverse time series domains, demonstrating the advantages of combining matrix profile techniques with deep learning for robust and scalable anomaly detection. Future work could focus on dynamic or multi-scale windowing techniques, adaptive thresholding methods, and testing in real-world applications like industrial control and financial monitoring to further enhance anomaly detection.

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

# A APPENDIX

## A.1 TIME COMPLEXITY ANALYSIS

Our method FMP-AE uses 1D-CNN to compute the Matrix Profile, which can significantly reduce the computational complexity compared to traditional methods. It can also leverage parallelism to accelerate execution and shorten runtime.

### (1) Traditional Matrix Profile Algorithms

The traditional Matrix Profile computation method uses a sliding window mechanism on the original time series to calculate the Euclidean distance or other similarity metrics between each window. Here we introduce two commonly used traditional algorithms. The first is **STAMP (Scalable Time series Anytime Matrix Profile)**, a brute-force global computation method that uses Fast Fourier Transform (FFT) to calculate the similarity between sliding windows, with a time complexity of $\mathcal{O}(n^2 \log n)$. The other is **STOMP (Scalable Time series Ordered Matrix Profile)**, an incremental update method that optimizes STAMP, reducing time complexity to $\mathcal{O}(n^2)$. However, it still operates at a quadratic complexity.

### (2) Our Method: FMP-AE

The 1D-CNN extracts local features through convolution operations. While convolution is still a sliding window operation, it reduces the computational load by sharing weights. The time complexity of CNN is generally $\mathcal{O}(nfk)$, where $n$ is the sequence length, $f$ is the number of filters, and $k$ is the kernel size. Comparatively, 1D-CNN can effectively lower the computational complexity, especially for long time series. Additionally, 1D-CNN operations are highly parallelizable, especially on GPUs or other hardware accelerators, which can significantly improve computational speed. Many Deep Learning Frameworks like PyTorch offer efficient GPU support for CNNs. By moving the model and data to the GPU, convolution operations and backpropagation are automatically parallelized. we use 'torch.cuda()' to ensure both the model and data run on the GPU. Convolution operations also are highly parallel because the computation for each output feature map is nearly independent, allowing for parallel processing of different windows and channels. On the other hand, we can process multiple time series segments at once, called Batch Processing. By increasing the $batch\_size$, data processing parallelism can be significantly improved, especially on multi-GPU setups. Specifically, each sequence segment in a batch can be independently processed, enabling parallel computation. Finally, for the computation of similarity matrices in the matrix profile, parallel matrix operation libraries (such as GPU tensor operations in PyTorch) can be utilized to accelerate the process. This enables the distribution of pairwise computations across multiple threads or GPU cores. Additionally, CNNs reduce feature redundancy between windows, further speeding up the computation. For instance, by selecting appropriate filter numbers and kernel sizes, the computational load of CNNs can be significantly reduced without substantially affecting performance.

Consequently, compared to traditional matrix profile algorithms, using 1D-CNN for feature extraction and MP computation reduces the complexity from $\mathcal{O}(n^2)$ to $\mathcal{O}(nfk)$, making it more efficient, particularly for long time series. 1D-CNN is highly parallelizable, especially on GPUs, where batch processing, convolution operations, and matrix computations can greatly accelerate both training and inference. Increasing the $batch\_size$, enabling GPU computation, parallelizing matrix operations, and leveraging multi-core CPUs are effective strategies to optimize parallelism. Our method substantially reduces the time required for matrix profile computation while effectively leveraging parallelism to accelerate the runtime. This advancement enables real-time or near-real-time anomaly detection in large-scale datasets, expanding its applicability to time series anomaly detection applications.

## A.2 LOSS FUNCTION

The loss function of our model is to combine the reconstruction error of the Autoencoder and the *MP* loss of the 1D-CNN feature map. It can be expressed as in the following:

$$\mathcal{L}_{\text{total}} = \tilde{\mathcal{L}}_{\text{recon}} + \lambda \cdot \tilde{\mathcal{L}}_{\text{MP}},$$

where, $\tilde{\mathcal{L}}_{\text{recon}}$ is the reconstruction error of the Autoencoder with Batch Normalization, measuring how effectively the model can reconstruct the input time series, $\tilde{\mathcal{L}}_{\text{MP}}$ is the difference in the matrix

profile between feature maps extracted by the 1D-CNN, also with Batch Normalization, $\lambda$ is a dynamically adjusted weighting parameter used to balance the reconstruction loss and the *MP* loss, increasing gradually during training.

This loss is based on the difference between the input data and the model's reconstruction, commonly measured using Mean Squared Error (MSE):

$$\mathcal{L}_{\text{reconstruction}} = \frac{1}{m} \sum_{i=1}^{m} (x_i - \hat{x}_i)^2 .$$

The second derivative (i.e., the Hessian matrix) is:

$$\frac{\partial^2 \mathcal{L}_{\text{reconstruction}}}{\partial \theta^2} = \frac{2}{m} \sum_{i=1}^{m} \frac{\partial^2 \hat{x}_i}{\partial \theta^2}$$

The second derivative of the loss function measures its curvature, which reflects the convergence behavior during optimization. In the case of MSE (Mean Squared Error), the second derivative is constant, providing stable curvature that facilitates convergence. MSE directly quantifies the Euclidean distance between the original and reconstructed time series, assessing whether the model has effectively captured the characteristics of normal data. For normal data points, the reconstruction error is expected to be small, whereas for anomalies, the error will be large. This makes MSE a reliable criterion for anomaly detection. Moreover, as a quadratic function with a positive definite Hessian matrix, MSE is a standard convex function. This ensures that the reconstruction error term is convex, guaranteeing that any local minimum is also a global minimum, which is crucial for stable and efficient optimization.

The Matrix Profile is generated by computing the feature similarity matrix of the time series. For feature representations $F$, the similarity matrix is computed as $S(i,j) = \text{similarity}(F_i, F_j)$. This similarity matrix reflects the similarity between different parts of the time series. After computing the similarity matrix, it is further optimized into a Matrix Profile, which provides a global view of time series similarity indicators, usually calculated via a sliding window. This similarity matrix reflects the similarity between different parts of the time series. After computing the similarity matrix, it is further optimized into a Matrix Profile, which provides a global view of time series similarity indicators, usually calculated via a sliding window. The convexity of the matrix profile loss $\|S - P\|$ depends on the choice of similarity measure. If Euclidean distance is used and the CNN feature extraction layers are designed appropriately, this loss function is also convex. Even if the non-linear structure of the CNN may introduce local non-convexity, the dynamically adjusted $\lambda$ helps to gradually overcome local minima traps.

**Batch Normalization** In our loss function design, batch normalization is applied to ensure consistency in loss calculations across different batches, addressing potential variations in data distribution. We normalize using the mean reconstruction error within each batch for the reconstruction loss, which effectively adjusts the scale of the loss. This normalization is based on the average reconstruction error across all batches, ensuring that reconstruction errors are comparable across batches. The Matrix Profile loss is normalized using the global mean of the Matrix Profile values across all batches. Batch normalization is necessary because batch-level data characteristics may vary significantly. Without normalization, it could lead to instability in the training process, such as gradient explosion or premature convergence. By normalizing based on actual reconstruction errors and Matrix Profile values, we ensure that each batch contributes more evenly to the total loss function, facilitating more effective model learning and improving overall training stability and robustness.

**Dynamically Adjusted Coefficient** $\lambda$ During model training, $\lambda$ is incrementally adjusted to increase the emphasis on matrix profile error over time. This approach offers several benefits. At the early stages of training, the model primarily focuses on reconstructing input data using the Autoencoder, which allows it to learn the overall structure of the data. As training advances, $\lambda$ is gradually increased, shifting the model's attention toward minimizing the matrix profile error. This encourages the model to focus more on local feature similarities or emerging anomaly patterns, which is crucial for detecting subtle irregularities in the time series.

This gradual increase in $\lambda$ is a strategic choice. If the model were to emphasize anomaly detection too early, it might converge prematurely to suboptimal solutions or fail to grasp the global structure

of the data. By initially prioritizing global feature learning and progressively incorporating the matrix profile error, the model can effectively capture both broad patterns and local anomalies.

**Gradient Clipping** In addition, during the optimization of the loss function, gradient clipping is employed to mitigate the risk of gradient explosion, a problem that becomes particularly critical in the later stages of training when $\lambda$ increases. As the matrix profile error term grows in influence, it can lead to larger gradient updates during backpropagation, potentially destabilizing the training process. Without proper control, these large gradients could cause abrupt changes in model weights, resulting in convergence issues or even complete divergence.

By applying gradient clipping, we limit the magnitude of the gradients to a predefined threshold, ensuring that updates remain within a stable range. This not only preserves the model's ability to learn from significant error signals but also prevents excessive weight shifts that could otherwise cause the loss function to oscillate or explode. Consequently, gradient clipping serves as an essential procedure, promoting stable convergence while allowing the model to effectively learn both global structures and local anomaly as $\lambda$ continues to increase during training.

In essence, gradient clipping strikes a balance between preventing overly aggressive updates and maintaining the flexibility needed for the model to adapt to the increasing emphasis on the matrix profile error, ensuring smoother and more reliable optimization.

### A.3 MODEL ARCHITECTURE AND HYPERPARAMETER

In this section, we examine the architecture of the 1D-CNN and Autoencoder models, listing details such as convolution kernel size, number of channels, activation functions, and pooling layers.

**(1) 1D-CNN**

The 1D-CNN has three Convolutional layers (Conv) followed by Batch Normalization(BN), Max Pooling layers, and two Fully Connected layers (FC). The input to this model is a one-dimensional time series with $window\_size$ as the input sequence length. The structure of 1D-CNN is shown in the following Table 3.

Table 3: Architecture of the 1D-CNN

| Layer | Type | Kernel Size | Stride | Padding | Output Channels | Activation | Pooling |
|-------|------|-------------|--------|---------|-----------------|------------|---------|
| Conv1 | Conv1d | 5 | 1 | 2 | 16 | ReLU | MaxPooling (kernel=2, stride=2) |
| BN1 | BatchNorm1d | – | – | —— | – | – | – |
| Conv2 | Conv1d | 5 | 1 | 2 | 32 | ReLU | MaxPooling (kernel=2, stride=2) |
| BN2 | BatchNorm1d | – | – | – | – | – | – |
| Conv3 | Conv1d | 3 | 1 | 1 | 64 | ReLU | MaxPooling (kernel=2, stride=2) |
| BN3 | BatchNorm1d | – | – | – | – | – | – |
| FC1 | Fully Connected | – | – | – | 128 | ReLU | Dropout (p=0.3) |
| FC2 | Fully Connected | – | – | – | 64 | – | – |

Next, we will provide the input and output dimension changes for each layer of the 1D-CNN model. Here, we assume that the input $window\_size$ is 128. The following Table.4 is a detailed description of each layer from input to output.

**(2) Autoencoder**

Table 4: Architecture of 1D-CNN with Input Size = (1, 128)

| Layer | Input Shape | Output Shape | Operation |
|---|---|---|---|
| Input | (1, 128) | (1, 128) | Raw time series input |
| Conv1 + ReLU | (1, 128) | (16, 128) | Conv1d with 16 filters (5x1) |
| MaxPool1 | (16, 128) | (16, 64) | MaxPool1d (kernel=2, stride=2) |
| Conv2 + ReLU | (16, 64) | (32, 64) | Conv1d with 32 filters (5x1) |
| MaxPool2 | (32, 64) | (32, 32) | MaxPool1d (kernel=2, stride=2) |
| Conv3 + ReLU | (32, 32) | (64, 32) | Conv1d with 64 filters (3x1) |
| MaxPool3 | (64, 32) | (64, 16) | MaxPool1d (kernel=2, stride=2) |
| Flatten | (64, 16) | (1024) | Flatten the feature map |
| FC1 + ReLU | (1024) | (128) | Fully connected layer (128 units) |
| Dropout | (128) | (128) | Dropout (p=0.3) |
| FC2 | (128) | (64) | Fully connected layer (64 units) |
| **Output** | **(64)** | | Feature vector output |

The Autoencoder consists of two fully connected layers for the encoder and two fully connected layers for the decoder. Assuming the input $window\_size$ is 128, the input is flattened to one dimension before passing through the Autoencoder. The hierarchy changes as shown in the Table.5.

Table 5: Architecture of the Autoencoder

| Layer | Type | Input Dimension | Output Dimension | Activation |
|---|---|---|---|---|
| Encoder FC1 | Fully Connected | Input (depends on window size) | Input/2 | ReLU |
| Encoder FC2 | Fully Connected | Input/2 | Input/4 | ReLU |
| Decoder FC1 | Fully Connected | Input/4 | Input/2 | ReLU |
| Decoder FC2 | Fully Connected | Input/2 | Input | Sigmoid |

We continue to assume an input $window\_size$ of 128, which is flattened before passing through the Autoencoder layers. The following Table.6 shows the layer-wise changes.

Table 6: Architecture of the Autoencoder with Input Size = 128

| Layer | Input Shape | Output Shape | Operation |
|---|---|---|---|
| Input | (1, 128) | (128) | Flatten the time series window |
| Encoder FC1 | (128) | (64) | Fully connected layer (128/2) |
| Encoder FC2 | (64) | (32) | Fully connected layer (64/2) |
| Decoder FC1 | (32) | (64) | Fully connected layer (32 to 64) |
| Decoder FC2 | (64) | (128) | Fully connected layer (64 to 128) |
| Output | (128) | (1, 128) | Reshape to original input shape |

**(3) Total Parameters**

Now, we calculate the total number of parameters for both models. The number of parameters in a convolutional layer is given by the formula:

$$\text{Number of Parameters} = (\text{Input Channels} \times \text{Kernel\_Size} + 1) \times \text{Output Channels}$$

where the "+1" accounts for the bias term.

- **Conv1**: $(1 \times 5 + 1) \times 16 = 96$ parameters.

- **Conv2**: $(16 \times 5 + 1) \times 32 = 2592$ parameters.
- **Conv3**: $(32 \times 3 + 1) \times 64 = 6208$ parameters.
- **FC1**: $1024 \times 128 + 128 = 131200$ parameters.
- **FC2**: $128 \times 64 + 64 = 8256$ parameters.

Thus, the **total number of parameters for the 1D-CNN** is: $96 + 2592 + 6208 + 131200 + 8256 = 148352$.

Then, we discuss the parameters of Autoencoder. The number of parameters in a fully connected layer is given by the formula:

$$\text{Number of Parameters} = (\text{Input Dimension} \times \text{Output Dimension}) + \text{Output Dimension}$$

When the input size is 128, the total number of parameters for the encoder and decoder are:

- **Encoder FC1**: $128 \times 64 + 64 = 8256$ parameters.
- **Encoder FC2**: $64 \times 32 + 32 = 2080$ parameters.
- **Decoder FC1**: $32 \times 64 + 64 = 2112$ parameters.
- **Decoder FC2**: $64 \times 128 + 128 = 8320$ parameters.

Thus, the **total number of parameters for the Autoencoder** is: $8256 + 2080 + 2112 + 8320 = 20768$.

- The **total parameters for 1D-CNN**: 148,352.
- The **total parameters for Autoencoder**: 20,768.
- The **total number of parameters for the entire model**:

$$148352 + 20768 = 169120$$

The entire model has approximately **169k** parameters, making it relatively compact and highly suitable for efficient anomaly detection in time series data. This streamlined architecture not only accelerates the training process but also reduces computational resource requirements. The compact design enhances the model's scalability, making it suitable for real-time applications as well as environments with limited computational power. Additionally, the reduced number of parameters helps mitigate the risk of overfitting, thereby improving the model's generalization ability to new data.

### A.4 FEATURE MAP EXTRACTED BY 1D-CNN

Anomaly detection is designed to identify inputs that deviate significantly from established data patterns. At its core, the model's task is to extract features that can distinguish between normal and anomalous instances. Convolutional layers within Convolutional Neural Networks (CNNs) excel at this task by automatically extracting local features from input data. These features might include abrupt changes, periodic variations, and other structural characteristics, all of which are critical for distinguishing between normal and anomalous patterns in time series or signal processing contexts. The essence of anomaly detection is to pinpoint rare events or phenomena that diverge notably from typical patterns. The convolutional layers' outputs play a crucial role in this process, as they provide essential insights into the presence of anomalies. Sharp changes, such as spikes in the feature maps, are often indicative of anomalies, while smooth or low-frequency variations generally reflect normal behavior.

When input data contains anomalies, these are often manifested as significant fluctuations or abnormal peaks in the outputs of specific convolutional filters. For instance, certain channels might exhibit more pronounced changes compared to others, highlighting anomalous regions within the input data. As illustrated in Figure 8, feature maps derived from normal data typically display predictable and repetitive patterns. In contrast, anomalies can cause certain channels to produce outputs that significantly deviate from the norm, showing up as spikes or abrupt changes. Smooth channel outputs, which are nearly constant, suggest that no significant features have been captured, indicating that the data may be normal. On the other hand, significant fluctuations in channel outputs often

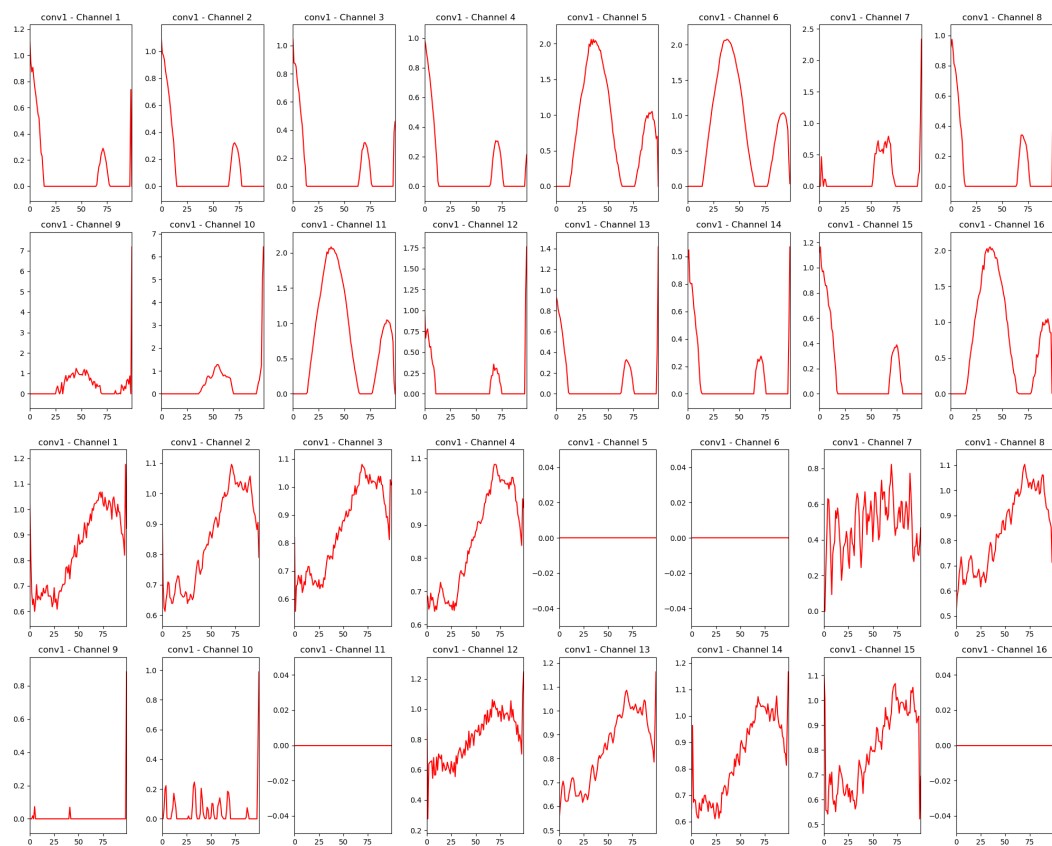

Figure 8: Feature maps extracted by 1D-CNN

point to anomalous patterns, especially when a convolutional filter produces sudden peaks over a specific time interval. Such sudden peaks frequently signal the presence of an anomaly.

By analyzing these convolutional layer outputs, one can effectively identify and understand anomalous patterns, thereby enhancing the accuracy and efficiency of anomaly detection systems. This approach ensures that both the subtle and dramatic deviations from normal patterns are captured and addressed appropriately.

**Feature Map & Matrix Profile** When calculating the Matrix Profile, larger values in the feature maps indicate that certain segments of the time series exhibit pronounced characteristics, suggesting these segments are unique. In contrast, smaller feature map values imply that these segments are relatively ordinary and do not stand out in the model. To compute the similarity between different time series windows, the features extracted by the 1D-CNN (i.e.,the feature maps) serve as the primary input.

The process begins with calculating the Euclidean distance between feature maps to construct a similarity matrix. This matrix captures the similarity between various time series windows by measuring how closely related they are in terms of their extracted features. Following this, the Matrix Profile is computed based on the minimum distances found in the similarity matrix. Smaller values in the Matrix Profile denote higher similarity between windows, suggesting that the segments are closely related. Conversely, larger values indicate significant differences between windows, which may help in identifying unusual or anomalous time segments.

Therefore, the Matrix Profile reflects both large and small values from the feature maps. Specifically, anomalies detected in the feature maps tend to result in higher values in the Matrix Profile, signaling potential anomalies or deviations from normal patterns. On the other hand, normal values in the feature maps lead to lower values in the Matrix Profile, indicating that the time series segment is similar to other segments and is more likely to represent typical, non-anomalous behavior. This

approach allows for a nuanced understanding of both normal and anomalous patterns in time series data.

A.5   MATRIX PROFILE FROM ORIGINAL TIME SERIES AND FEATURE MAP

In this section, we will compare the matrix profile calculated using traditional methods on the raw time series with which computed using our model.

Stumpy is an efficient Python library designed specifically for time series analysis. Its core function is the computation of the Matrix Profile, a critical tool for time series pattern discovery, similarity search, and anomaly detection.

**Length Difference**  As shown in the Figure 9 below, the Matrix Profile extracted by 1D-CNN is more **shorter**. This is mainly because 1D-CNN reduces the original time series through pooling layers and convolution operations, producing feature vectors that are shorter than the original input. This reduction helps decrease computational complexity while retaining important features. In traditional Matrix Profile methods, the similarity of each subsequence is computed individually, so the length of the output Matrix Profile is generally equal to the length of the original time series minus the length of the subsequence. In contrast, 1D-CNN significantly reduces the data resolution through its convolutional kernels and pooling layers, resulting in a shorter Matrix Profile. Below, we will compare the advantages of using the 1D-CNN model versus the Matrix Profile computed with the Stumpy library for anomaly detection by analyzing Figure 9.

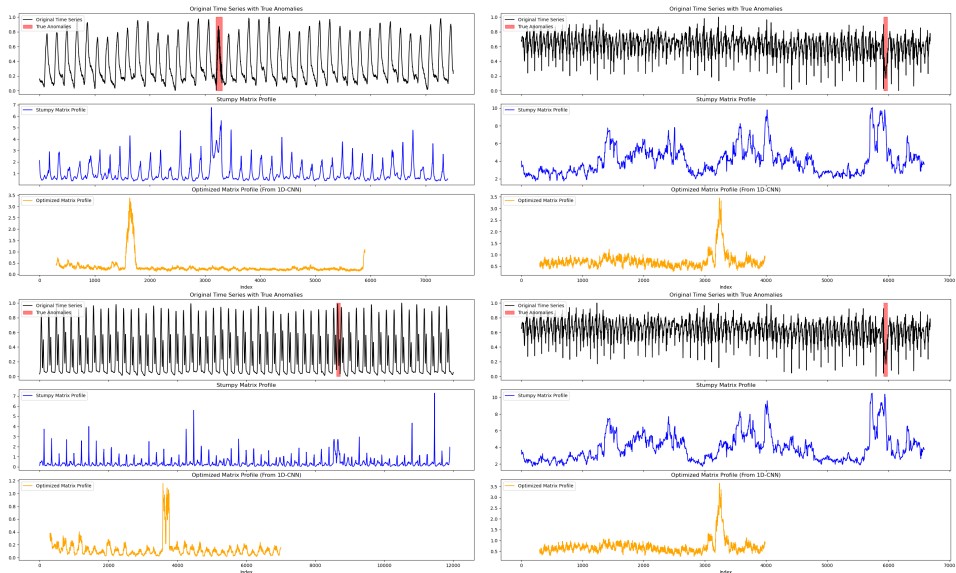

Figure 9: The Stump calculating Matrix profile and 1D-CNN calculating Matrix profile

**(1) Clarity of Anomalies and Peak Distribution**

In the second part of each figure, the results from Stumpy show multiple peaks with relatively frequent fluctuations. Although these fluctuations might indicate potential anomalies, their frequent occurrence complicates threshold setting. Multiple peaks suggest that several subsequences within the time series have low similarity, which can lead to false positives and obscure the true anomalies. Additionally, when compared to the ground truth (the first part of each figure), it is evident that each sequence contains only one anomaly, which is a characteristic of the UCR250 datasets. In contrast, the results from our model present anomalies that are more concentrated and distinct. The figure reveals only a single significant global peak, indicating that the model successfully identifies one anomaly while other areas remain relatively low. This clear peak demonstrates that our model more accurately captures anomalies without generating false positives from irrelevant fluctuations. In other words, it effectively reduces false positives, thereby increasing precision.

**(2) Global vs Local Perspective**

Since the Stumpy library uses a sliding window technique based on Euclidean distance for its calculations, it primarily identifies patterns through local similarities. When the time series contains multiple similar patterns or noise signals, Stumpy's results often exhibit numerous local peaks. Not all of these peaks may correspond to actual anomalies, which complicates the process of setting thresholds and accurately identifying anomalies. In contrast, our model captures data features from a broader, global perspective through advanced feature extraction mechanisms. This approach allows the model to filter out some of the local noise and focus on identifying global patterns. As a result, our model is more attuned to overall trends, concentrating on genuine anomalies while maintaining stable outputs in other regions. This makes the task of setting thresholds more straightforward and effective.

**(3) Noise Filtering and Model Robustness**

While the Stumpy method is efficient, it is relatively sensitive to noise and can easily interpret minor fluctuations and local deviations in the data as anomalies, leading to multiple peaks. Our model, through convolutional kernels and subsequent layers that integrate these features, is potentially more robust and capable of filtering out minor noise. The orange curve in the figure shows smoother anomaly detection results with minimal significant noise, aside from actual anomalies. This smoothness indicates that the model has strong noise resistance and effectively focuses on the main trends and true anomalies in the time series.

**(4) Reliability and Efficiency of Anomaly Detection**

The peaks calculated using Stumpy, while representing some local anomalies, may require manual intervention to adjust detection thresholds due to their numerous and uneven distribution. For applications, this results in lower reliability because the complexity of the peaks increases the risk of false positives and missed detections. The 1D-CNN model, however, significantly simplifies the anomaly detection process. With its very clear and concentrated output, it is easy to set a higher detection threshold, focusing only on the prominent global peaks, thus maintaining a high detection rate while reducing false positives.

Therefore, we conclude that our model has these advantages over the Stumpy-based Matrix Profile in anomaly detection:

- Anomalies are more prominent and concentrated, reducing the likelihood of false positives.

- Threshold setting is simpler, without the need to handle multiple local peaks

- Greater robustness, capable of filtering out noise and focusing solely on global anomalies.

- Smoother output makes the model more stable and reliable, suitable for various real-world anomaly detection scenarios.

