# OpenReview forum: "FMP-AE: A HYBRID APPROACH TO TIME SERIES ANOMALY DETECTION"
_ICLR.cc/2025/Conference — Submitted to ICLR 2025_

### Official Review · Reviewer_FbFe · 2024-10-20

**Soundness:** 3
**Presentation:** 3
**Contribution:** 3
**Rating:** 5
**Confidence:** 4

**Summary:**

This paper proposed a model based on the computation of Feature map Matrix Profile, combined with an Autoencoder (FMP-AE) for time series anomaly detection. It introduces a novel loss function which enhances the model’s ability to detect anomalies by leveraging both global similarity and local feature reconstruction. The Matrix Profile is also designed based on the 1D-CNN extracted feature for a rapid and reliable identification of anomalies. Extensive experiments on UCR250 benchmark datasets demonstrate the effectiveness of the proposed method.

**Strengths:**

- The unsupervised time series anomaly detection task is quite challenging, the author build a unified hybrid solution and demonstrate great results.

- The paper is well-organized and clearly written, and it provides enough figures and demonstration to explain the methods well.

- I like the ablation study part, it clearly shows the effectiveness of each utilized component.

**Weaknesses:**

The paper writing and the proposed method all look good to me, however, I do have concerns on the experiment part.

- The experiments are only conducted on the UCR Anomaly Detection dataset, while the recent papers usually conduct experiments on multiple datasets to make the results more convincing (e.g., apart from UCR, AD Transformer also reports the results on SMD , PSM, MSL& SMAP, SWaT,  NeurIPS-TS etc.).

- The compared methods are relatively old (AD transformer is proposed on 2021 and published on 2022), how about compared with some recent  methods (2023 ~ 2024). For example,

[D3R]  Drift doesn’t matter: Dynamic decomposition with diffusion reconstruction for unstable multivariate time series anomaly detection

[GPT4TS] One fits all: Power general time series analysis by pretrained LM

[ModernTCN] ModernTCN: A modern pure convolution structure for general time series analysis.

[SensitiveHUE]  Sensitivehue: Multivariate time series anomaly detection by enhancing the sensitivity to normal patterns.

- It's hard to say it outperforms ADTransformer on UCR, since the recall of UCR results is higher, there's a trade of between the precision and the recall (F1 measurement usually cannot accurately catch this trade off). Besides, from their paper, the P=72.80, R=99.60, and F1=84.12, which are different from what you reported.

**Questions:**

- Did you also evaluate on other datasets? Could you share some data points?

- It would be better if you also compared with the recent methods.

- Could you provide some info on why ADTransformer is different from their reported results, and what might be the cause?

---

> ### Author Response · Authors · 2024-11-24
> **Responses to comments of Revoewer FbFe**
>
> We thank Reviewer FbFe for his positive feedback and for giving us valuable comments and suggestions, which helped us improve the manuscript's quality. Some more detailed responses are given below.
>
> Responses to Weaknesses and Questions 1. The UCR250 dataset contains 250 time series across 12 domains, including healthcare, finance, and engineering. Unlike other mentioned datasets, it consists of univariate time series to which our method is devoted. Extending the method to multivariate data needs more dedicated study and more experiments. This will be a good direction for our future work.
>
> Responses to Weaknesses and Questions 2. Thanks to the reviewer for this suggestion. We incorporated the 2023 DC detector method into our experiments and compared the FMP-AE model with it. Our model outperforms the DC detector in terms of precision and F1 score. The other mentioned methods treat the case of multivariate time series, and their papers have no experiments on the UCR250 dataset.
>
> Responses to Weaknesses and Questions 3. Perhaps due to the adjustment of some hyperparameters, we have not fully run the results presented by ADtransformer on our own device. In the revised manuscript, we will change the experimental results to those provided by the original authors.

---

> > ### Comment · Reviewer_FbFe · 2024-12-01
> >
> > Thanks for the reply, but I will insist my score since the responses didn't address my concern well.

---

### Official Review · Reviewer_C5e5 · 2024-10-30

**Soundness:** 2
**Presentation:** 3
**Contribution:** 1
**Rating:** 3
**Confidence:** 4

**Summary:**

This paper claims that there are three challenges facing by time series anomaly detection work: 1) label scarcity; 2) generalizability; 3) efficiency. Based on this recognition, the authors proposes a time series anomaly detection method by combining a matrix profile and an auto-encoder. The authors believe the matrix profile can improve the method efficiency, but the other two challenges was not discussed. The authors have made some experiments to verify the effectiveness of their method, but not compared with the most SOTA baselines. Besides, they also do not prove that the proposed method can improve method efficiency experimentally.

**Strengths:**

1. Overall well written. The paper is easy to follow and understand.
2. The idea of matrix profile is somewhat novel in time series anomaly detection.

**Weaknesses:**

1. Unsolved challenges. As mentioned in summary part, this paper mainly proposes three challenges. However, the authors only claim that the proposed method is more efficient by introducing matrix profile. The other two challenges is not solved by the proposed method. Also, it is not clear why matrix profile combining with an auto-encoder can improve the method efficiency compared with a solely auto-encoder.
2. Unpersuasive experiments. On one hand, the proposed methods do not compare with the most SOTA methods, for example [1] [2] [3]. The latest method included in baseline is published in 2022. On the other hand, since the authors claimed that introducing matrix profile can improve method efficiency in their contributions, they should also have made experiment to prove it. However, this aspect is not discussed thoroughly in experiment.
3. Limited contribution. There are mainly two part in the proposed method: one is matrix profile and another is an auto-encoder. The structure of auto-encoder is really common by combining max-pool and CNN. The matrix profile may be somewhat novel, but it is a really marginal improvement.
4. Misunderstanding of the shortage of deep-learning-based anomaly detection methods. In the abstract, the authors claimed that deep-learning-based anomaly detection methods is affected by data imbalance problem. Actually, many unsupervised deep-learning-based anomaly detection methods assume there are fewer anomalous samples in training set, so that when training by it, the model can better fit the normal pattern rather than anomalous one. Thus, deep-learning-based anomaly detection method actually benefit from those data imbalance.

[1] Yang Y, Zhang C, Zhou T, et al. Dcdetector: Dual attention contrastive representation learning for time series anomaly detection[C]//Proceedings of the 29th ACM SIGKDD Conference on Knowledge Discovery and Data Mining. 2023: 3033-3045.

[2] Li Y, Chen W, Chen B, et al. Prototype-oriented unsupervised anomaly detection for multivariate time series[C]//International Conference on Machine Learning. PMLR, 2023: 19407-19424.

[3] Tuli S, Casale G, Jennings N R. Tranad: Deep transformer networks for anomaly detection in multivariate time series data[J]. arXiv preprint arXiv:2201.07284, 2022.

**Questions:**

1. Could you please add some statics about the proposed method efficiency and baseline efficiency?
2. Why matrix profile combining an auto-encoder can improve the method efficiency, compared with a solely auto-encoder?

---

> ### Author Response · Authors · 2024-11-24
> **Responses to Reviewer C5e5**
>
> We thank Reviewer C5e5 for giving us valuable comments and suggestions, which helped us improve the manuscript's quality. Detailed responses are given below.
>
> Responses to W1. The matrix profile improves efficiency by focusing on data similarity and temporal relationships, reducing noise. Combining it with the autoencoder enhances feature extraction and better uses unlabeled data when labels are scarce. It also helps reveal data patterns and anomalies, reducing reliance on labeled data and improving task generalization. We will improve our manuscript to make it clearer.
>
> Responses to W2. Thanks to the reviewer for this suggestion. We incorporated the 2023 DC detector method (cf. [1]) into our experiments and compared the FMP-AE model with it. Our model outperforms the DC detector in terms of precision and F1-score. The UCR250 dataset contains 250 univariate time series across 12 domains, including healthcare, finance, and engineering. The UCR250 dataset consists of univariate time series to which this model is devoted. The other two mentioned methods treat the case of multivariate time series, and their papers have no experiments on the UCR250 dataset. Extending our method to multivariate data needs more dedicated study and more experiments. This will be a good direction for our future work.
>
> Responses to W3. While the improvement may seem modest, using high-dimensional features from CNN enhances anomaly detection by reducing noise and irrelevant patterns. Our method combines traditional matrix profiles with deep learning, offering a novel approach. The key innovation lies in integrating the optimized matrix profile with an autoencoder, achieving superior performance on UCR250. To our knowledge, this combination is new.
>
> Responses to W4. We agree that many unsupervised deep-learning-based methods assume fewer anomalies in the training set, allowing the model to fit normal patterns better. Our statement in the abstract did not mean to suggest that data imbalance always hinders performance. Instead, we aimed to highlight that extreme imbalance can still pose challenges, such as distinguishing subtle anomalies or handling overlapping distributions. To clarify, we will revise the abstract to reflect this nuance, such as: "While deep-learning-based anomaly detection methods often benefit from data imbalance by focusing on learning normal patterns, extreme imbalances or subtle anomalies can still pose challenges."

---

> > ### Comment · Reviewer_C5e5 · 2024-11-25
> >
> > **Response to "Response to W1"**
> >
> > You should use experiment or theoretically proof to prove it, rather than just say it by word. Also, the authors claimed that the matrix profile help to reveal the data patterns and reduce the reliance on labeled data, but the method is still a supervising one, which will not reduce the labels needed.
> >
> > **Response to "Response to W2"**
> > Thank you for supplement an experiment. However, it have not proven that using matrix profile can improve method efficiency yet.
> >
> > **Response to "Response to W3""**
> >
> > The combination is new. However, it is really a limited innovation.
> >
> > **Response to "Response to W4"**
> >
> > Thank you for your further elaboration.

---

> > > ### Author Response · Authors · 2024-11-25
> > > **Response to Reviwer C5e5**
> > >
> > > Thank you for taking the time to read our response and paper.
> > > In fact, about the supervised learning task question, we did not use any training set labels during the training process; we only used the raw data. Labels were only used during the testing phase to compute evaluation metrics. Therefore, our study remains an unsupervised learning task.
> > > Additionally, our ablation experiments have demonstrated the effectiveness of Matrix Profile for anomaly detection in univariate time series.

---

> > > > ### Comment · Reviewer_C5e5 · 2024-11-26
> > > >
> > > > Thank you for your further clarification.
> > > > At present stage, I think the proposed method can solve the problem of label scarcity. However, whether it can solve the problem of algorithm efficiency and generality remains unknown, after all, there is no experimental or mathematical proof can prove this.
> > > > By the way, the ablation experiment can prove the effectiveness of matrix profile, but I mean the efficiency of matrix profile, that is how much the matrix profile can increase the algorithm speed.

---

> > > > > ### Author Response · Authors · 2024-11-26
> > > > > **Response to Reviwer C5e5**
> > > > >
> > > > > Thank you for your valuable feedback. Regarding the improvement in algorithm efficiency, our approach is based on calculating the Matrix Profile of the feature map after feature extraction (Section 3.1), which we refer to as the optimized Matrix Profile. Compared to directly computing the Matrix Profile on the raw time series, this method significantly reduces computational complexity while still effectively extracting anomaly features. Therefore, we believe this approach not only enhances the model's efficiency but also ensures its effectiveness.

---

> > > > > > ### Comment · Reviewer_C5e5 · 2024-11-26
> > > > > >
> > > > > > I mean as the efficiency is a main contribution highlighted by your paper, it should be proved by experiment rather than just using some words to give some explaination.

---

### Official Review · Reviewer_VVFZ · 2024-10-31

**Soundness:** 1
**Presentation:** 3
**Contribution:** 1
**Rating:** 3
**Confidence:** 4

**Summary:**

The paper proposes a combination of CNN and matrix profile to detect anomalies. The paper is easy to follow.

**Strengths:**

1. The paper studies an important problem.
2. It is interesting to consider matrix profile in deep learning.

**Weaknesses:**

1. The challenges presented in the paper are well known and have been solved by many other time series anomaly detection methods, such as label scarcity. The paper lacks an argument on why existing methods fails to address these challenges and why the proposed method show advantages over the existing methods to solve these challenges.
2. The method is quite easy and straightforward. It uses CNN to extract features of time series, and uses MP to compute similarity between subsequences. These are well-known techiniques.
3. The baselines and datasets are limited. SOTA baselines, such as DCdetector, ModernTCN, D3R, etc., are missing. Only UCR dataset is used, and many other well-known datasets are missing, such as MSL, PSM, SMAP, etc.

**Questions:**

See weaknesses.

---

> ### Author Response · Authors · 2024-11-24
> **Responses to comments of Reviewer VVFZ**
>
> We thank the Reviewer VVFZ for giving us valuable comments and suggestions, which help us improve the manuscript's quality. Some more detailed responses are given below.
>
> Responses to W1. Although challenges in time series anomaly detection are well known, and there exist some methods to tackle them, there are still improvements to be done in terms of Precision, Recall, and F1 score.
> Compared to these methods, our method has the advantage of running faster and more efficiently. We will consider giving more descriptions in terms of the efficiency of time series anomaly detection.
>
> Responses to W2. In fact, we did not directly calculate the MP between the subsequences but calculated the MP of the features extracted by CNN, which is different from the traditional MP method, and theoretically, it is an "optimized MP.” We will describe the method in more detail.
>
> Responses to W3. Thanks to the reviewer for this suggestion. We incorporated the 2023 DC detector method into our experiments and compared the FMP-AE model with it. Our model outperforms the DC detector in terms of precision and F1-score. The UCR250 dataset contains 250 univariate time series across 12 domains, including healthcare, finance, and engineering. Unlike datasets MSL, PSM, SMAP, the UCR250 dataset consists of univariate time series to which this model is devoted. The other two mentioned methods treat the case of multivariate time series, and their papers have no experiments on the UCR250 dataset. Extending our method to multivariate data needs more dedicated study and more experiments. This will be a good direction for our future work.

---

> > ### Comment · Reviewer_VVFZ · 2024-11-27
> >
> > Thank you for your reply. However, I don't feel the rebuttal fully addressed my concerns, such as novelty, technical depth, and comparison with baselines. Thus, I will maintain my score.

---

### Official Review · Reviewer_r5k6 · 2024-11-02

**Soundness:** 2
**Presentation:** 2
**Contribution:** 1
**Rating:** 3
**Confidence:** 5

**Summary:**

The author proposes a new method for time-series anomaly detection, FMP-AE, which combines the TCN network with the MP algorithm to achieve better detection performance.

**Strengths:**

The writing is clear and easy to understand.

**Weaknesses:**

Insufficient innovation. The organization of the paper is not good enough and the experiments are not sufficient.

**Questions:**

1. The organization of the introduction has a lot of room for improvement. The author's focus should be on TSAD (Time-Series Anomaly Detection) rather than general anomaly detection. Therefore, the introduction should be revised. Additionally, it should explain why past methods face challenges and what their deficiencies are.
2. The ablation experiment lacks analysis of 1D-CNN + AE.
3. The analysis in the ablation experiment part lacks some more valuable conclusions. It should not be just a simple comparison of the performance of different methods.
4. Figure 4 does not show the original time series and abnormal regions, making it difficult to understand the pros and cons of different losses. Moreover, necessary analysis is lacking.
5. Figures 5 and 6 lack analysis, making it impossible to understand the analysis results of Figure 5. The significance of Figure 6 is unclear, and it is impossible to know the analyzed object and results.
6. The analysis conclusion of Figure 7 seems to overlap with the previous ones and lacks more insightful conclusions. At the same time, there is no comparison with other methods, lacking persuasiveness.
7. Figures 4-7 do not indicate on what data the experiments are conducted and there is no comparison with other methods.
8. In setting up comparison methods, recent works such as NPSR, SimAD, D3R, LLM and other models are lacking.
9. Figure 10 is difficult to understand, with a large amount of blank time series.
10. The author does not analyze from the experimental and theoretical levels why their method can solve existing challenges.
11. There is a lack of necessary comparison of space-time complexity and algorithm time consumption, making it impossible to effectively prove that the author's proposed algorithm is efficient.
12. The author only validates on one dataset, UCR, lacking mainstream datasets in the current TSAD community such as MSL, SMAP, SMD, etc. It is difficult to evaluate the performance of the algorithm under complex conditions.
13. There is a lack of analysis of the classic Matrix Profile algorithm. What would be the performance if features are obtained using TCN or MLP and then the classic MP is used?
14. The author's experimental organization is not good enough and does not effectively verify the motivation of their algorithm.
15. I suggest that the author open source the code for follow-up research.

---

> ### Author Response · Authors · 2024-11-24
> **Responses to comments of Reviewer r5k6 Part 1**
>
> We thank the Reviewer r5k6 for giving us valuable comments and suggestions, which helped us improve the manuscript's quality. Detailed responses are given below.
>
> Response to Question 1. Thank you for the insightful feedback. In response, we have revised the introduction to focus explicitly on Time-Series Anomaly Detection (TSAD), emphasizing its unique challenges—such as label scarcity, generalizability across domains, and computational efficiency—while addressing the limitations of existing methods in these areas. We have also reorganized the introduction to clearly define TSAD, outline the shortcomings of past approaches, and highlight how our proposed FMP-AE model, with its novel Matrix Profile loss and efficient 1D-CNN-based design, effectively tackles these issues. We believe these changes significantly improve the manuscript.
>
> Response to Question 2. Thank you for the valuable feedback. We have added an analysis of the 1D-CNN + AE experiment. In this setup, the model achieves 90.45% accuracy and an F1-score of 69.30%. While 1D-CNN and Autoencoder effectively capture local features and reconstruct data, the absence of MP loss weakens the model's precision and overall performance. This demonstrates the importance of MP loss in improving detection accuracy and balancing precision and recall.
>
> Response to Question 3. We have enriched the ablation analysis by emphasizing the insights gained from each experiment, such as the critical role of 1D-CNN in capturing local features, the importance of the MP loss in enhancing precision, and the complementary contributions of Autoencoder and 1D-CNN. These findings highlight how each component addresses specific challenges in time-series anomaly detection, offering a deeper understanding beyond performance comparisons.
>
> Response to Question 4. We have updated Figure 4 to include the original time series and annotated anomalies, along with added analysis comparing the strengths and limitations of reconstruction loss and MP loss in detecting anomalies.
>
> Response to Question 5. We have added a more detailed analysis for Figures 5 and 6. For Figure 5, we now explain the relationship between the original and detected anomalies, highlighting how the model performs on validation data. For Figure 6, we clarify the significance of the AUC-ROC curve, discussing how it reflects the model's performance in terms of true positive and false positive rates at various thresholds. These changes should make the analysis and the results more comprehensible and meaningful.
>
> Response to Question 6. Thank you for the valuable feedback. We have removed Figure 7 in our revised manuscript to avoid any overlap with previous figures.
>
> Response to Question 7. We have updated the figure captions to specify that the experiments are conducted on validation data. Additionally, we have included a comparison with other methods in the tables to better contextualize the results and highlight the effectiveness of our approach. These updates should provide clearer context and improve the comprehensiveness of the analysis.
>
> Response to Question 8. Thanks to the reviewer for this suggestion. We incorporated the 2023 DC detector method into our experiments and compared the FMP-AE model with it. Our model outperforms the DC detector in terms of precision and F1 score. The other mentioned methods treat the case of multivariate time series, and their papers have no experiments on the UCR250 dataset.
>
> Response to Question 9. Figure 10 shows the feature maps extracted by the CNN. In some regions, there may be no significant features, resulting in blank areas, but this does not imply that the original time series is blank.
>
> Response to Question 10. We have revised the theoretical analysis in Section 3.1. We also emphasize the significance of our comparative experiments and ablation studies more explicitly, highlighting how the hybrid FMP-AE method effectively integrates traditional matrix profile techniques with deep learning and outperforms existing methods in key performance metrics. These changes help to strengthen the clarity and impact of our conclusion.

---

> > ### Author Response · Authors · 2024-11-24
> > **Responses to comments of Reviewer r5k6 Part 2**
> >
> > Response to Question 11. Thank you for your valuable feedback. As with most of the studies in the domain, we have included an analysis of the time complexity in Appendix 1.
> >
> > Response to Question 12. The UCR250 dataset contains 250 time series across 12 domains, including healthcare, finance, and engineering. Unlike the other mentioned datasets, we chose this dataset because it consists of univariate time series to which our method is devoted. Extending the method to multivariate data needs more dedicated study and more experiments.
> >
> > Response to Question 13. We appreciate the suggestion to analyze the performance of the classic Matrix Profile algorithm when features are obtained using TCN or MLP. While we have focused on integrating Matrix Profile with deep learning techniques in our proposed method, we acknowledge that this comparison could provide additional insights. However, given the scope of our current work, we have chosen not to include this analysis at this stage. We will, however, consider it for future research to explore further the impact of different feature extraction methods on Matrix Profile performance.
> >
> > Response to Question 14. Thank you for the feedback. We appreciate your concern regarding the experimental organization. In response, we have already adjusted the experimental setup to align more effectively with the motivation behind our algorithm. These changes include refining the experimental design to demonstrate better how our approach addresses the challenges outlined in the introduction. We believe these modifications strengthen the validation of our method's effectiveness.
> >
> > Response to Question 15. Thank you for the suggestion. We appreciate your interest in our work and fully agree on the importance of open-sourcing code for follow-up research. We plan to upload the code to an anonymous GitHub repository and put the link in the revised manuscript.

---

> > > ### Comment · Reviewer_r5k6 · 2024-11-28
> > >
> > > Thank you for your response and the revisions to the article, but I still insist on my score. The article still has significant shortcomings in both experimentation and textual logic.

---

### Official Review · Reviewer_ZEhb · 2024-11-03

**Soundness:** 1
**Presentation:** 2
**Contribution:** 1
**Rating:** 5
**Confidence:** 4

**Summary:**

This paper presents FMP-AE, a hybrid model for unsupervised anomaly detection in time series, combining Matrix Profile (MP) structures with deep learning components, specifically a 1D Convolutional Neural Network (1D-CNN) and an Autoencoder. The novelty lies in introducing an MP-based loss function that complements the Autoencoder’s reconstruction loss to improve anomaly detection performance. The model is evaluated on the UCR250 benchmark, where it demonstrates strong performance across several metrics.

**Strengths:**

1. Combines Matrix Profile with Deep Learning for Anomaly Detection: The paper’s main contribution lies in the integration of Matrix Profile with an Autoencoder, aiming to leverage both global and local sequence information for more robust anomaly detection.

2. Positive Empirical Results on UCR250 Dataset: FMP-AE shows strong empirical results on the UCR250 dataset, indicating that the model is capable of achieving high precision, recall, and F1 scores within this dataset, and the authors conduct ablation studies to explore the contribution of each model component.

**Weaknesses:**

1. Incremental Methodological Innovation: While combining Matrix Profile (MP) with Autoencoder reconstruction loss is practical, it mainly builds on existing methods without substantial theoretical innovation. The A.2 Loss Function section explains how this combination may enhance anomaly detection but lacks deeper theoretical justification for why MP loss would improve sensitivity beyond reconstruction loss alone. The explanation is largely empirical, without rigorous theoretical support to clarify this enhancement mechanism.

2. Restricted Dataset Evaluation and Generalizability Concerns: The evaluation is limited to a single dataset, UCR250, which restricts insights into FMP-AE’s applicability across diverse real-world scenarios. Furthermore, there is no theoretical justification for the combined loss function, making it unclear whether the observed improvements generalize beyond this dataset. Expanding the evaluation to include more varied datasets or providing theoretical analysis could help validate the model’s robustness and adaptability.

3. Comparison with Outdated Baselines: The baseline methods used for comparison are primarily from 2022 or earlier, omitting more recent approaches in anomaly detection. This limits the assessment of FMP-AE’s competitiveness against the latest advancements in the field, making it difficult to gauge how the proposed method stands relative to state-of-the-art techniques. A comparison with newer methods would provide a clearer picture of FMP-AE’s effectiveness within the current landscape.

**Questions:**

1.	Could the authors offer further theoretical justification for the combination of MP loss with reconstruction loss? Specifically, what theoretical basis supports the idea that MP loss enhances anomaly detection sensitivity beyond what reconstruction loss alone can achieve?

2.	How does FMP-AE perform on other datasets beyond UCR250, and how generalizable is the model across different domains? Additional experiments on varied time series data would help clarify the model’s robustness and adaptability.

3.	Why were no post-2022 methods included in the baseline comparisons? Would a comparison with more recent techniques clarify FMP-AE’s standing within the current anomaly detection landscape?

---

> ### Author Response · Authors · 2024-11-24
> **Responses to comments of Reviewer ZEhb**
>
> We thank the Reviewer ZEhb for giving us valuable comments and suggestions, which helped us improve the manuscript's quality. Some more detailed responses are given below.
>
> 1. Responses to W1 and Q1. We respect the reviewer's opinion on theoretical justification and rigorous theoretical support. However, this reproach can be made to most of the studies in the area. We have made a theoretical introduction of the loss function in Appendix A.2. The theoretical basis is that the reconstruction loss pays more attention to the global information of the time series, so it may ignore the detailed extraction of local information. After using 1D-CNN to extract information, the local anomaly information captured by MP loss can be retained to a greater extent.
>
> 2. Responses to W2 and Q2. The UCR250 dataset contains 250 time series across 12 domains, including healthcare, finance, and engineering. Unlike SMD, MSL, and SWAT, we chose this dataset because it consists of a univariate time series to which this method is devoted. Extending the method to multivariate data needs more dedicated study and more experiments. This will be a good direction for our future work.
>
> 3. Responses to W3 and Q3. Thanks to the reviewer for this suggestion. We incorporated the 2023 DC detector method into our experiments and compared the FMP-AE model with it. Our model outperforms the DC detector in terms of precision and F1-score.

---

### Meta-Review · Area_Chair_eG4F · 2024-12-18

**Metareview:**

The work tackles the problem of unsupervised anomaly detection on time series data and introduces a method that enhances matrix profiling with autoencoding for handling several issues in the problem, including data imbalance, generalization, and efficiency issues.

**Strengths.**
- The idea of enhancing matrix profiling with autoencoder is new.
- The method shows effective performance on UCR250 datasets.
- The paper is generally well written.

**Weaknesses.**
- The proposed method lacks technical novelty.
- The experimental justification is not convincing due to, e.g., the lack of up-to-date competing methods, more diverse datasets, etc. Major claims are not properly justified, such as the claim on efficiency.
- The motivation and the justification of the model design are found to be insufficient.
- All five reviewers generally agree on the above three weaknesses.

**Additional Comments On Reviewer Discussion:**

Four out of five reviewers participated in the author-reviewer discussion. Unfortunately, they all believe that the rebuttal does not address their major concerns. Major concerns that are raised again by the reviewers during discussion, including on technical novelty and depth, comparison with baselines, efficiency justification, clarity of the method and the experiments. Consequently, no reviewers change their primary ratings, including four rejects and one weak reject. I agree with the reviewers that the paper is not ready for publication at ICLR.

---

### Decision · Program_Chairs · 2025-01-22

Reject